# Diet Digestibility and Partitioning of Nutrients in Adult Male Alpacas Fed a Blend of Oat Hay and Alfalfa Pellets at Two Levels of Intake

**DOI:** 10.3390/ani13233613

**Published:** 2023-11-22

**Authors:** Paola Katherine Chipa Guillen, Walter Orestes Antezana Julián, Wilfredo Manuel Rios Rado, Juan Elmer Moscoso-Muñoz, Edward H. Cabezas-Garcia

**Affiliations:** 1Facultad de Agronomía y Zootecnia, Universidad Nacional de San Antonio Abad del Cusco, Av. de La Cultura 733, Cusco 921, Peru; 150580@unsaac.edu.pe (W.M.R.R.); juan.moscoso@unsaac.edu.pe (J.E.M.-M.); 2Department of Animal Science, Cornell University, Ithaca, NY 14853, USA; ec867@cornell.edu

**Keywords:** alpacas, diet digestibility, feeding levels, maintenance, nutrient balance

## Abstract

**Simple Summary:**

Despite of the importance of alpacas to the economy of rural communities in the Peruvian Andes, studies on animal energetics and the protein metabolism of these animals are particularly lacking. In this study, a high-quality diet consisting of a blend of oat hay and alfalfa pellets (70:30 ratio as a percentage on a fed basis) was offered to male Huacaya alpacas, simulating two levels of intake (separate experiments): a maintenance level and an *ad libitum* level (40 and 50 g of DM per kg of BW^0.75^, respectively), using metabolism crates. The apparent organic matter digestibility and partitioning of nutrients (energy, C, and N) were rather similar for both experiments, despite the increased fecal output at the *ad libitum* level of intake.

**Abstract:**

Alpacas are well adapted to consume the poor-quality forages present in the arid conditions of the Andean Altiplano. However, studies focusing on understanding the relationship between diet digestibility and intake are scarce. The aim of this study was to evaluate the effects of two levels of dry matter intake (DMI) on a metabolic body weight (BW^0.75^) basis. The effects of a maintenance level of intake and an *ad libitum* level of intake (40 and 50 g of dry matter (DM) per kg of BW^0.75^, respectively) on the apparent diet digestibility and partitioning of specific nutrients (energy, carbon (C), and nitrogen (N)) of alpacas that were fed a blend of oat hay and alfalfa pellets (70:30 ratio as a percentage on a fed basis) were evaluated. Five adult intact male alpacas (BW = 62.9 ± 8.09 kg at the beginning of the study) were fed with the experimental diet and trained to be allocated in metabolism crates for 30 days. After the completion of this phase, two separate experiments for each level of intake were carried out, each lasting for twenty-six days (with the final five days taken for samples and raw data collection). In both experiments, the animals responded differently in terms of nutrient supply and changes in BW (−140 and 100 g/d for the maintenance and *ad libitum* levels of intake, respectively). Oat hay consumption was rather similar in both experiments, which may be explained by a reduced ingredient selectivity at the *ad libitum* level of intake. Diet digestibility was similar in both experiments, despite the greater fecal output of nutrients with the increased level of diet intake. In line with this, diet metabolizability, calculated as the ratio between metabolizable energy (ME) and gross energy (GE) contents, indicated a similar energy utilization of the diet. The apparent digestibility of the organic matter (OMD) ranged from 655 to 669 g/kg DM. Water consumption at the *ad libitum* level of intake was 21% higher than the observed mean at the maintenance level of intake. Fecal outputs of dietary energy, C, and N accounted for the largest source of excreted nutrients, regardless of the level of intake. The N retention increased from 0.439 at the maintenance level of intake, to 0.473 g of DM/kg BW^0.75^ when the alpacas were fed *ad libitum*.

## 1. Introduction

According to the Peruvian Institute of Statistics and Informatics [1], Peru has the largest inventory of alpacas worldwide, with 5.7 million of heads in 2022, making it the leading population among the domesticated South American camelid (SAC) species in the country. Alpacas are important for maintaining the health of the Andes ecosystems and are culturally significant for its rural communities. They also play a major role in providing food and serving as a source of net income through the processing of fleece for the textile industry [2,3,4]. Despite this, alpacas are still raised mostly in harsh conditions from the environmental point of view and with a low degree of technology [2]; as a result, low fertility [5] and reduced meat and fiber production [2] have been documented.

The challenging environmental conditions present in the Puna tussock rangelands (≥3000 m of altitude) represent a serious constraint for establishing any livestock production, including ruminants. In addition to factors such as the intense UV radiation, poor soil quality, etc., the rainy season can be as short as only three months in length, usually from December to March (annual precipitation can be as low as ≤1000 mm), and the environmental temperature can drop up to −10 °C during the dry season [6,7]. All these factors combined contribute to reduced forage production and the poor-quality of native grasslands during most parts of the year.

South American camelids have major anatomical and physiological differences when compared to ruminants, which enable them to be better adapted to survive the arid conditions of the Andean Altiplano [8,9]. The features of the digestive physiology of SAC are as follows: 1. a lower feed intake per unit of metabolic weight with an increased retention time of the digesta in their gastrointestinal tract, 2. an increased rate of absorption of volatile fatty acids (VFA’s) from the fermentation of dietary carbohydrates, 3. greater efficiency in the recycling of urea N via saliva, and 4. enhanced microbial protein synthesis [6,10,11,12]. Although in most aspects, the gastric anatomy of alpacas resembles that of llamas, the differences in the proportions of the stomach and the intestines are explainable when considering the different feeding behaviors of both species [13]. Llamas are better adapted to coarse forages [14], whereas alpacas select a wider variety of forage types compared to llamas [9]. Up to 47 plant species have been identified to be part of the diet of adult alpacas in the Peruvian Altiplano, with *Festuca dolichophylla* (grass), *Eleochoris albibracteata* (sedge), and *Achemilla pinnata* (forb) being their main forage choices [15].

Despite the reduced nutritional quality of the Peruvian Altiplano rangelands during the dry season [15,16,17], earlier studies have shown that adult alpacas counterbalance this situation by increasing their intake of *Festuca dolichophylla*/*Muhlenbergia fastigiata* swards up to 12% when compared to their observed intake during the rainy season [18]. This is explained by the observed increase in the gastric capacity of these animals in response to the reduced forage quality [11]. Dry matter intake (DMI) and diet digestibility are the two main factors driving production responses in farm animals and that is not an exception with alpacas. However, little is known about digestibility and the partitioning of nutrients, especially when alpacas are fed at the maintenance level of intake.

Considering the difficulties associated with providing a continuous supply of native forages for conducting a balance study with alpacas caged in metabolism crates, previous studies have used conventional roughages (e.g., oat hay, fresh ryegrass) to mimic the nutritional quality of native forages [18]. Moreover, balance studies are useful to unveil feed digestion mechanisms in herbivores [19,20,21]. In the present study, two separate experiments were conducted to investigate the relationship between two feeding levels and the utilization of nutrients in alpacas. The main hypothesis is that an increased DMI of a high-quality diet will enable alpacas to utilize dietary energy and nutrients (C and N) more efficiently.

## 2. Materials and Methods

### 2.1. Animals, Management, and Data Collection

A nutritional balance study was carried out between November 2019 and January 2020 at the Maranganí Research Station of the Veterinary Institute for Tropical and High-Altitude Research (IVITA Maranganí) belonging to Universidad Nacional Mayor de San Marcos (UNMSM) in Cusco, Peru. The geographical coordinates of IVITA Maranganí are 14°21′24.51″ S (latitude) and 71°10′4.34″ W (longitude). This research station is located at 3704 m above the sea level. Five intact Huacaya male alpacas (body weight; BW: 62.9 ± 8.09 kg; 4.5 years old, on average at the beginning of the study) were used as experimental animals. These males were obtained from an extensive production system on native grasslands dominated by *Festuca* sp., *Calamagrostis* sp., and *Alchemilla* sp. Upon the arrival of the alpacas to the research facility, a complete veterinary examination, including weighting and deworming, were performed for each individual animal to guarantee their good health status.

Thirty days prior to the commencement of the study (pre-experimental period), caretakers gradually trained the alpacas to become accustomed to experimental routines and to be individually allocated in metabolism crates (dimensions: 2.0 m × 0.55 m × 2.2 m). The five stainless steel crates were equipped with a ramp for the easy access and withdrawal of the animals following the standard model for herbivores, as proposed by Milne et al. [20]. The optimal size of these cages was previously determined using the measurements of seven morphometric variables performed on 30 alpacas belonging to the IVITA Maranganí herd. In addition, the operation and validation aspects of using these metabolism crates, such as animal welfare and cleanliness, easy collection of feed and water refusals, and evaluation of the alpacas’ performance when fed at the maintenance level of intake, were previously performed, and further details can be found in our previous article [22]. During this time period, the animals were fed twice a day (07:00 and 13:00 h) at the maintenance level of intake (40 g of DM per kg of BW^0.75^), and water was provided *ad libitum* using bowl drinkers. The experimental diet comprised a blend of oat hay and alfalfa pellets (70:30 ratio as a percentage on a fed basis) and its nutritional composition is presented in Table 1. Oat hay was chopped using a mini chaff cutter to obtain a mean particle size of 3.0 cm. Every second day and before feeding times, the experimental animals were released from the metabolism crates in a designated area in intervals varying from 10 to 15 min for physical exercise and to prevent locomotion problems [23]. These time intervals were also used for cleaning the metabolism crates and emptying both feeders and drinkers. During the morning time interval (07:00 h), BW was recorded using an electronic scale with a sensitivity of ± 0.5 kg (Tru-TestTM-Econo Plus; Auckland, NZ, USA). Recommendations provided by Lund et al. [24] were implemented prior to and during the experimental periods to minimize unnecessary animal stress.

Upon the completion of the pre-experimental phase, two separate experiments, each lasting 26 days, were carried out. Within these time periods, the first 21 days corresponded to the adaptation phase, and the last five days were taken for samples and data collection of feed and water intakes, diet refusals, feces, and urine, respectively. The aforementioned routines were kept during the experimental periods. Two levels of intake of the experimental diet, were offered to the animals; namely, a maintenance level of intake and an *ad libitum* level of intake (40 and 50 g of DM per kg of BW^0.75^, respectively). The comparative slaughter study by Roque H. et al. [27] was taken as a basis for establishing the feeding levels offered to the animals in the present study. Feed refusals were collected and weighted just before the morning feeding at 06:00 h, and feed intake (g/d) was calculated by difference. Water consumption (L/d) was also recorded by calculating their intake throughout the day, and an additional bowl drinker per cage (similar to the one already included in the metabolism crate) was used to correct for evaporation losses.

Measurements of daily fecal output (g/d) were performed by using sampling collection bags with a harness system attached to the perineal area of the animals, as shown by Vélez-Marroquín et al. [22]. Routinely, these bags were placed on the alpacas at 06:30 h, before morning feeding and after emptying fecal contents collected during the day. Fecal production was weighed and recorded for each animal daily. Urine was collected by gravity in a receptacle placed underneath the metabolism crates. A solution of 15 mL containing 10% sulfuric acid was added to the urine to prevent its deterioration due to microbial activity [28]. Representative samples (≈10%) of diet, feces, and urine collected over the five consecutive days were pooled for each individual animal and subsequently labelled and stored at −20 °C prior to chemical composition analyses.

### 2.2. Processing of Samples and Analyses of Chemical Composition

Samples of feeds, feed refusals, and feces were oven dried (60 °C) during 48 h for the initial determination of their DM contents and subsequently grounded in a knife mill using a 1-mm screen sieve (KN 295 Knifetec™, Foss, Hillerød, Denmark). Urine samples were lyophilized prior to the analyses. All chemical analyses were conducted in duplicate, and the majority of these were performed according to the official AOAC methods [29]. The residual moisture was determined using an oven drying method for 12 h at 105 °C and the DM contents of the samples were adjusted (method 950.46). Wet chemistry analyses were performed to determine ash contents (method 942.05), and both carbon (C) and nitrogen (N) contents were assessed according to method 990.03. Crude protein (CP) was calculated as N × 6.25. A neutral detergent fiber analysis [30] was performed with a heat-stable α-amylase and sodium sulfite using the filter bag technique in an Ankom200 fiber analyzer (ANKOM Technology Corp., Macedon, NY, USA), and expressed inclusive of residual ash (aNDF) [25]. The gross energy (GE) analysis of feed, fecal, and urine samples were performed using a Parr 6400 Oxygen Bomb Calorimeter following the recommended procedures provided by the manufacturer (Parr Instrument Co., Moline, IL, USA).

### 2.3. Diet Digestibility, Partitioning of Nutrients, and Estimation of Enteric CH_4_ Emissions

The apparent total-tract digestibility of dietary nutrients (DM, OM, CP, aNDF, and GE) was calculated from the feed intake and total fecal collection according to Cochran and Galyean [31]:
$$\mathrm{Apparent}\;\mathrm{total}\;\mathrm{tract}\;\mathrm{digestibility}\;=\frac{\mathrm{nutrient}\;\mathrm{intake}-\mathrm{fecal}\;\mathrm{output}}{\mathrm{nutrient}\;\mathrm{intake}}\times 1000$$
where, apparent total-tract digestibility is expressed in g/kg, and both nutrient intake and fecal output are expressed in g/d. Both C and N balances were assessed by a factorial approach, discounting for losses of specific nutrients where applicable. The absorbed (digested) nutrients were calculated by discounting fecal outputs from the initial intakes, and the retained nutrients were computed as the difference between the absorbed nutrients and those excreted in urine. The metabolizable energy intake (MEI) was calculated as the difference between the GE intake and energy losses in the feces, urine, and estimated enteric methane (CH_4_) production (g/d).

Since the present study was not originally intended for measuring enteric CH_4_ emissions, an empirical equation was used instead. By doing this, it was possible to obtain a complete calculation of the daily energy partitioning, and thus the MEI could be estimated. This equation was derived from unpublished data of studies conducted in 2021 at ‘La Raya’ Research Station (Universidad Nacional San Antonio Abad del Cusco, UNSAAC), under similar experimental conditions. Both the animal type and diet (including feeding levels) were the same as those evaluated in the present study. Daily CH_4_ emissions from the ‘La Raya’ studies were obtained from a head hood system over a period of 24 h for each individual animal, as described by Fernández et al. [32]. This equipment is unique in South America, and it has been customized for measuring the gas exchange and energy metabolism of SAC by taking advantage of its operational flexibility and affordable cost (C. Fernández, Universitat Politècnica de València, personal communication, 18 January 2023). The reliability of the CH_4_ measurements obtained from this head hood system at ‘La Raya’ conditions was previously evaluated, and additional information can be found in our recent validation study [33].

### 2.4. Statistical Analysis

The experimental data were analyzed by descriptive statistics (means and standard deviations) for each feeding level (maintenance and *ad libitum*) using the MEANS procedure of SAS (version 9.4; SAS Institute Inc., Cary, NC, USA). A linear mixed regression model for predicting the enteric CH_4_ emissions was derived from the ‘La Raya’ dataset by using the MIXED procedures of SAS, with the effect of individual alpacas considered in the RANDOM statement of the model. The obtained equation was as follows:
$$\mathrm{Enteric}\;CH_{4}=9.97(2.393)+0.476(0.2255)\times \mathrm{GEI}$$
where CH_4_ is the estimated enteric methane emission in grams per day, and GEI is the daily gross energy intake in MJ/d. The following factors were used in converting units: 1 g = 1.40 L = 55.5 kJ; 1 L = 0.716 g = 39.54 kJ [34,35].

## 3. Results

### 3.1. Animal Performance and Diet Digestibility

Table 2 presents the results for feed and water consumption, as well as the differences in BW. Overall, the average diet intakes on a metabolic BW basis were lower than the aimed levels (−6.5 and −20% for maintenance and *ad libitum* levels, respectively) and a greater variability was observed for the *ad libitum* level of intake (c.v. = 4.9 vs. 12%). The differences in BW indicated weight loss (−100 g/d) and gain (140 g/d) when the diet was offered at 40 and 50 g/kg of BW^0.75^, respectively. Oat hay intake was similar in both experiments, which is consistent with the decreased ingredient selectivity that was observed when the animals were fed *ad libitum*. Water consumption related to BW increased up to 19% with the increased level of intake. The results for fecal output, urine production, and apparent total-tract digestibility are presented in Table 3. As expected, fecal outputs of DM, OM, CP, and aNDF increased when the diet was offered *ad libitum*. Diet digestibility was rather similar in both experiments. The apparent OM digestibility (OMD) of the experimental diet was 669 and 655 g/kg of DM for the maintenance and *ad libitum* levels of intake, respectively.

### 3.2. Energy Partitioning and Balances of Nutrients

The daily energy partitioning and balances of nutrients (C and N) are presented in Table 4. The excretion of nutrients in feces was positively related to an increased level of intake. Alpacas fed at the maintenance level excreted 217 ± 37.7 kJ/kg of BW^0.75^, which represents 63% of the total GEI. Energy, C, and N contents in the urine did not differ among treatments; these contents were consistent with the total urine production. The estimated *Y_m_* (the fraction of dietary GE that is converted to CH_4_) was lower with an increased level of intake (4.59 ± 0.270 vs. 4.42 ± 0.424 as a % of the GEI, respectively). The sum of N excreted in both feces and urine accounted for 43% of the total N intake on average. Rather similar C:N ratios were observed for the maintenance and *ad libitum* levels of intake (21.6 and 21.4, respectively). However, the C:N ratios in the excreted nutrients (feces and urine) were slightly higher when the diet was offered *ad libitum*.

## 4. Discussion

Research that is focused on assessing nutrient utilization efficiency in farm animals at various levels of intake provides valuables insights for formulating appropriate nutritional recommendations. Traditionally, both energy and protein requirements of SAC are, in many cases, derived from those established for domestic ruminants [36,37], overlooking significant differences in their digestion mechanisms [6,11]. The nutritional quality of the diet is another important aspect to be considered. For example, in diets containing less than 7.5% CP, digestibility was found to be higher for alpacas than sheep, whereas no differences between species were observed in diets with levels higher than 10.5% CP [6]. This finding confirms that nutritional recommendations originally set for small ruminants are valid for alpacas only if they are fed with high-quality diets.

It is important to mention that the present study was carried out in two separate experiments for evaluating the same diet at maintenance (40 g of DM/kg BW^0.75^) and at *ad libitum* levels of intake (50 g of DM/kg BW^0.75^), respectively. Even though both experiments were conducted with the same animals, data cannot be analyzed as a change-over design, since the effects of both periods and the order of applying treatments for each individual animal were not included within the statistical analysis, as all five alpacas concurrently received the same level of intake [38]. Because of this, we decided to analyze the data using descriptive statistics of two separate experiments. We do believe that the data presented in our study are valuable for comparison purposes, which is confirmed by the findings in the literature discussed in the following sections.

### 4.1. Body Weight Changes and Nutrients Intakes

Due to differences related to the experimental conditions of the present study, discrepancies may arise with the literature reports in BW changes, especially at the maintenance level of intake (expected to be as closer as possible to zero BW gain). For example, in the study by Roque H. et al. [27], growing male alpacas (two-year-old) that were offered a diet consisting of oat hay and alfalfa hay (50:50 on a fed basis) at the same intake level as the maintenance level of intake in the present study (40 g of DM/kg BW^0.75^), underwent a BW loss of −14.6 g/d. In the present study, adult male alpacas that were fed at the targeted maintenance level of intake lost 100 g/d, and substantial variations among animals were observed. This value is contrary to the value reported by Vélez-Marroquín et al. [22] for the same type of animals and diet, but with a slightly higher maintenance intake (42.7 g of DM/kg BW^0.75^; 73.3 g/d).

The intakes of oat hay were rather similar at the maintenance and *ad libitum* levels (572 and 599 g/d, respectively). These values were aligned with a decreased ingredient selectivity in the *ad libitum* level (91 vs. 76%). However, this was not reflected in large differences in the aNDF intake of the total diet, which in turn encompassed observed trends for the other nutrients. As it has been well documented for domestic ruminants, fiber contents in the diet control the DMI capacity of SAC, but to a lesser extent (0.8–0.9% of BW) due to the slower passage rate of digesta [7]. The intake potential driven by the aNDF contents in the diet was almost achieved at the maintenance level of intake (0.75% of BW). The relatively high content of aNDF of the oat hay that was offered to the alpacas in the present study (625 g/kg DM) might have contributed to C1 (compartment 1) fill as a limiting factor. This might explain the modest increase in DMI when the experimental diet was offered *ad libitum* (from 37.4 ± 1.83 to 40.0 ± 4.89 g of DM/kg BW^0.75^). In this respect, the study conducted by Obregón Cruz [39] found that offering alpacas oat hay *ad libitum* (NDF content of 621 g/kg DM) led to a decreased DMI (39.4 g of DM/kg BW^0.75^).

The detrimental effect of fiber contents in the forage on DMI responses is very well illustrated in the study conducted by Paredes et al. [40], in which alpacas that were fed *ad libitum* received four contrasting levels of NDF in the diet by changing the proportions of stems and leaves in the oat hay. Although in the present study the intake potential of the experimental alpacas was not constrained by the CP content in the diet (126 g/kg DM), it is possible that the increased consumption of alfalfa pellets at the *ad libitum* level of intake contributed to an increased redox potential in C1, which in turn might have led to the reduction in fiber digestion [41]. In order to understand the effect of the aNDF content in the diet on alpaca DMI in more detail, it is recommended to further examine this relationship in future studies.

The observed DMI at the maintenance level (37.4 g of DM/kg BW^0.75^) is in line with the results reported by Huareccallo Maquera [42] and Roque H. et al. [27] for growing animals: 36.7 and 37.4 g of DM/kg BW^0.75^, respectively. According to Bonavia and McGregor [43], the DMI required for alpacas that are fed a diet comprised of alfalfa hay and whole barley grain at the maintenance level is 41 g of DM/kg BW^0.75^ (1.48% of BW). The values found in the present study are consistent with values previously reported by Hoffman and Fowler [44], who used different feedstuffs, and who reported a DMI at maintenance level ranging from 1.25 to 1.50 as a % of BW. The observed DMI in the present study at the level of 50 g of DM/kg BW^0.75^ with adult animals was 1.41% of BW or 40.0 g of DM/kg BW^0.75^. These values were slightly lower than those reported by Roque H. et al. [27] for growing alpacas.

Comparative studies have shown that SAC, including alpacas, consume less food than sheep [9,45]. In line with this, San Martin and Bryant [6] reviewed data from six studies conducted under confinement conditions with a variety of feeds (*n* = 16) offered *ad libitum* and concluded that the DMI, expressed as a percentage of BW, was on average 20% higher for sheep than alpacas (2.29 vs. 1.83% of BW, respectively). As a result of the lower energy requirement for maintenance, a lower DMI can be expected in SAC when compared to ruminants in general [46]. In the study conducted by Roque H. et al. [27], two-year-old male alpacas achieved a maximum intake of 1.92% of BW (or 56.4 g of DM/kg BW^0.75^) when a diet of up to 70 g of DM/kg BW^0.75^ was offered, whereas in the balance study conducted on Australian mature alpacas by Liu et al. [41], the authors found the DMI to be equivalent to 1.77% of BW when the animals received a diet containing alfalfa hay and concentrate *ad libitum* (78:22 ratio on a DM basis).

As it occurs in ruminant animals, the chemical composition of the diet is a major driver of feed intake responses in alpacas. For example, in a case where four-year-old alpacas in the Central Region of Chile were offered either wheat straw or a high-quality ryegrass hay, the *ad libitum* DMI was as low as 39.9 g of DM/kg BW^0.75^ and as high as 63.1 g of DM/kg BW^0.75^ [47]. In the present study, despite the level of diet intake offered to the experimental animals, the observed levels of water consumption meet the recommendations for SAC in controlled environments, as recommended by Rübsamen and Engelhardt [48] and Van Saun [46].

### 4.2. Apparent Digestibility of the Diet

The apparent digestibility of nutrients (DM, OM, CP, aNDF, and GE) were rather similar despite the level of DMI. Although this trend is consistent with findings by Huareccallo Maquera [42] and Roque H. et al. [27], the average DM digestibility (DMD) values reported in those studies were considerably lower (609 and 614 g/kg, respectively) when compared with the increased diet digestibility in our study. This difference is attributable to an increased fecal production as a percentage of the total DMI in the studies of Huareccallo Maquera [42] and Roque H. et al. [27] Conversely, the DMD reported by Vélez-Marroquín et al. [22], who administered the same diet as the one offered in the present study, is slightly higher than the value found in the present study at the maintenance level (668 g/kg DM for the referred study). Therefore, it appears that the alfalfa pellets (offered at 30% on a feed basis) likely have contributed to increased diet digestibility when compared to more fibrous diets [27,42].

In the study by Robinson et al. [23], when three roughages with contrasting CP contents were offered to intact male alpacas, as the CP contents in the roughage increased, a greater DMD was observed: wheat straw: 259 g/kg, tall fescue hay: 619 g/kg, and alfalfa hay: 639 g/kg. In the same study, this trend was also replicated for N digestibility: 408, 681, and 786 g/kg for the same roughages, respectively.

Similar OMD values as those found in the present study (662 g/kg OMD on average for both levels of intake) have been reported by Liu et al. [41] regardless of the diet offered to the animals (sorghum sudan, alfalfa hay, and fresh alfalfa diets, each of them with the fixed inclusion of 160 g of concentrate/alpaca per day). In addition to the CP contents of the roughages per se, Liu et al. [41] hypothesized that differences in the degradability of dietary protein fractions may explain the decreased CP digestibility (CPD) observed in alpacas with the alfalfa hay diet vs. the fresh alfalfa diet (648 vs. 748 g/kg DM, respectively). In the present study, the mean apparent CPD’s of the diet were: 746 ± 45.7 and 728 ± 35.8 g/kg DM for the maintenance and *ad libitum* levels of intake, respectively.

Differences in cell wall digestion can be attributed to intrinsic factors determining the potential digestibility of NDF in ideal rumen conditions and to extrinsic animal and diet-related factors (e.g., intake level, diet composition) that determine the extent of the intrinsic digestibility achieved [49]. In the present study, the fecal excretion of aNDF increased when the alpacas were fed *ad libitum*, as expected. However, this was not reflected in a substantial decrease in the apparent aNDF digestibility of the diet. Likely, it appears that neither the CP supply nor digesta retention time (linked to feeding level) constrained the fiber digestion performed by microorganisms present in C1, as it is well established that the intrinsic chemical characteristics of the fiber offered in the diet is an important aspect to be considered. The study by López et al. [47] showed that the digestibility of the NDF was greater for alpacas that were fed wheat straw and fescue *ad libitum* (543 g/kg DM on average) than for those that were fed clover (469 g/kg DM; greater in lignin contents). In the same study, DMI was positively correlated with DMD (r = 0.52), which in turn was highly and positively correlated with NDF digestibility (r= 0.88).

The aforementioned correlations for alpacas that are fed low-quality forages, are contrary to what has been observed for dairy cows and sheep fed different forage-to-concentrate ratios, where increases in DMI have been associated with digestibility depression [50]. The same trend was observed by Dias et al. [51] with beef steers that were fed oat hay at restricted and *ad libitum* levels of intake. Conversely, the apparent digestibility rates of NDF were greater for alpacas that were fed sorghum sudan diets (542 g/kg DM) than for those that were fed alfalfa hay and fresh alfalfa diets (453 and 433 g/kg DM, respectively) according to Liu et al. [41]. The comparative study by Sponheimer et al. [12] demonstrates that alpacas showed a higher digestive efficiency than goats when offered a C_4_ grass hay (*Cynodon dactylon*), which is possibly due to the longer retention time of digesta particulate matter in their gastrointestinal tracts (71 and 54 h, respectively). However, no difference was found between these species when the animals were fed a C_3_ grass hay (*Bromus inermis*). Overall, based on the fecal outputs and apparent digestibility of the nutrients found in the present study for alpacas that were fed a high-quality diet, it seems that, as it has been observed in ruminants, the undigested OM can influence the rate of digestion [52]. It is also expected that both pools of fecal microbial N and endogenous N increase proportionally in response to the intake of fermentable OM [53].

The urine production was similar for both levels of diet intake (429 and 445 mL/d for maintenance and *ad libitum* levels, respectively), and it was equivalent to 19.3% of the total water consumption on average. These results broadly agree with the calculated value from tabulated data in Vélez-Marroquín et al. [22]. Previous studies have mentioned that the digestive efficiency of SAC increases at high altitudes [6,23,36]. It is possible that a connection exists between the high altitude where the present study was conducted (3704 m above the sea level) and the relatively high apparent digestibility of nutrients found.

### 4.3. Energy Partitioning and Estimated CH_4_ Emissions

In the present study, the energy requirement for maintenance (ME_m_) of an adult male alpaca was estimated to be 347 ± 52.0 kJ/kg of BW^0.75^ per day. This value broadly agrees to the value calculated from tabulated data by Roque H. et al. [27] in their comparative slaughter study (347 kJ/kg of BW^0.75^ per day). In farm animals, the total energy requirements differ depending on physiological stage, level of physical activity, and the energy content of the diet. The feeding level did not influence the diet metabolizability (*q_m_*), calculated as ME/GE (0.53; AFRC, 1993 [54]), which indicates a similar energy utilization of the diet.

Regardless of the feeding regime, the energy in the feces accounted for the greatest loss in energy partitioning of the diet, which accounted on average for 35% of the total GEI. These values are very consistent with those calculated by Liu et al. [41] for alpacas that were fed alfalfa hay diet *ad libitum* (35%), and those estimated from the studies by Huareccallo Maquera [42] and Roque H. et al. [27]: 36 and 39%, respectively. Despite that the apparent digestibility of the GE (GED) are similar to the numbers reported by Bonavia and McGregor [43] for alpacas that were fed oat hay diets, the fecal energy losses in that Australian study, were only equivalent to 17.5% of the GEI. The energy contents in the urine of the alpacas in the above-mentioned Peruvian studies did not differ among the targeted levels of intake, and this was also observed in the present study. However, the daily excretion of urinary energy may differ greatly depending on the type of diet offered to the alpacas fed *ad libitum*, as previously shown by Liu et al. [41].

As expected, the estimated total CH_4_ production increased with the feeding level, but the opposite was observed when the emission values were expressed as a % of the GE intake (*Y_m_*); this confirms that the CH_4_ yield is inversely related to the rate of passage of digesta, which has been observed in ruminants [55,56,57] and confined alpacas at high altitude conditions [27,42]. For a better understanding of this relationship, additional analyses of the C1 microbiome communities that are linked to VFA’s fermentation profile should be considered in future studies. The total CH_4_ emissions at both the maintenance and *ad libitum* levels of intake of the present study were estimated to be 17.0 ± 0.67 and 17.6 ± 1.27 g/d, with this representing 4.59 and 4.42% of the total GEI (*Y_m_*), respectively. For the same levels of intake but with younger animals, data retrieved from Huareccallo Maquera [42] were as follows: 17.4 and 23.3 g/d, for both 40 and 50 g of DM/kg BW^0.75^ treatments (*Y_m_* values: 6.35 and 7.65, respectively). Without considering the variations associated with either the CH_4_ measurement technique or the physiological condition of the animals, these numbers may suggest a strong effect of the type of diet on *Y_m_* emission values in alpacas.

For *ad libitum* feeding conditions, a Swiss study conducted by Dittmann et al. [58] indicated that for a female alpaca (53 kg; four-year-old) that was fed a diet consisting of alfalfa hay with a fixed amount of alfalfa pellets (53% of DMI), the total CH_4_ emission was 23.8 g/d (converted from L/d). On the other hand, Pinares-Patiño et al. [59] showed that depending on diet quality, the total CH_4_ emissions of alpacas might not be different when compared to sheep. This was the case for the *Y_m_* values when both animal species were fed indoors on alfalfa hay. However, for the same CH_4_ trait, alpacas emitted more than sheep under grazing conditions when they were fed either perennial ryegrass/white clover or birdsfoot trefoil pastures. In the present study, the rate of CH_4_ production (g) per unit of intake of digestible NDF (DNDFI; kg DM) was almost the same for both the maintenance and *ad libitum* levels of intake (91 g/kg on average). This value is similar to the numbers reported by Pinares-Patiño et al. [59], who fed both alpacas and sheep alfalfa hay at *ad libitum* levels. The linear relationship between DNDFI and CH_4_ production when alfalfa hay and perennial ryegrass/white clover data were pooled together was stronger (R^2^ = 0.76) in the study of Pinares-Patiño et al. [59], than the value found in the present study on adult male alpacas fed at 50 g of DM/kg BW^0.75^ (R^2^ = 0.44; not shown). This may indicate that the digestion of other nutrients (e.g., CP) was important for accounting for the estimated CH_4_ production in our study.

### 4.4. Carbon and Nitrogen Balances

Overall, the excretion of C in both feces and urine encompassed the observed trends in the partition of dietary energy. One of the aims of the present study was to evaluate the effect of the feeding level on dietary N utilization. Depending on the plant species, the CP contents of native grasses in the Peruvian Altiplano can decrease up to (or even below) 40 g/kg DM during the dry season [16]. In the study by Huareccallo Maquera [42], the total N intake was calculated to be 11.7 and 13.0 g/d for the 40 and 50 g of DM/kg BW^0.75^ feeding levels, respectively (CP of the diet: 92 g/kg DM). In the present study, the N intakes (g/d) at the maintenance and overfeeding levels of intake were 17.4 and 19.0 g/d, respectively (CP of the diet: 126 g/kg DM). According to the formulas included within the nutritional recommendations by Carmalt [60], the maintenance CP requirement for the alpacas used in the present study should be equal to 13.1 g/d. When the N intakes are expressed on a digestible basis (absorbed), the requirements should be 0.571 and 0.603 g/kg BW^0.75^, for the maintenance and *ad libitum* levels of intake, respectively. The digestible N requirement at the maintenance level set in the study by Huasasquiche [61], 0.381 g/kg BW^0.75^ (or on a CP basis, 2.38 g/kg BW^0.75^), is equivalent to 67% of the value found in the present study, with a high-quality diet offered at 40 g of DM/kg BW^0.75^.

Regardless of the level of diet intake, N retention accounted for 57% of the total N intake on average. This value represents more than two-fold of the estimated values from the data reported by Robinson et al. [23] for tall fescue hay (25%) and alfalfa hay (22%), respectively, or even greater when compared to the data reported by Davies et al. [62], either for barley (6.6%) or barley-alfalfa hay diets (5.1%). In the study by Liu et al. [41], the N retention calculated from their tabulated values are for sorghum sudan diets (43%), fresh alfalfa diets (41%), and alfalfa hay diets (28%), respectively. The comparative slaughter study by Condori Apaza [63] evaluated the N efficiency of alpacas at similar experimental conditions as those described by Huareccallo Maquera [42] and Roque H. et al. [27]. In the study of Condori Apaza, for growing male alpacas (BW = 40 kg; two-year-old), the N retention accounted for 58 and 40% of the total N intake when the diet was offered at 40 and 50 g of DM/kg BW^0.75^, respectively. Thus, at the same maintenance level of intake, the first value is in concordance with the one found in the present study. The observed discrepancy at the 50 g of DM/kg BW^0.75^ level is likely explained by differences in excreted N in both feces and urine (not reported by Condori Apaza [63]).

Excreted N in feces increased with the level of diet intake. The available N, calculated as the N retained divided by the N absorbed, did not differ among treatments, which indicates that a high amount of N from the diet was apparently being absorbed by the gut (79% on average). Robinson et al. [23] compared the effect of feeding three forages of varying quality *ad libitum* to intact male alpacas of four age groups and found the same value as the one observed in the present study when the animals were offered an alfalfa hay diet (CP = 160 g/kg DM), whereas for a diet of tall fescue hay (CP = 118 g/kg DM), the available N decreased up to 67%. In the same study, when the animals were fed wheat straw (CP = 40 g/kg DM), this value decreased significantly up to 32%, and resulted in a negative N balance.

In the present study, the N excreted in the urine was very low (16% of the total N intake when diet was offered *ad libitum*) compared with the reported values in the previously aforementioned N balance studies (Davies et al. [62]; Liu et al. [41]; and Robinson et al. [23]). Among these, the closest value to the observed values in the present study was found by Liu et al. [41] for the alpacas that were fed a sorghum sudan diet, which was two-fold (32%), whereas for the alpacas that were fed an alfalfa hay diet, Robinson et al. [23] reported a value as high as 72%. The discrepancies in the urinary N contents may arise from increased urine production (ml/d) in the literature studies. Unfortunately, the referred N balance studies did not report results for this variable. On the other hand, studies conducted in the Peruvian regions of Puno [64,65] and Cusco [22] showed similar values to the values observed in the present study.

Based on the results of the C:N ratios in the feces and urine, in conjunction with a high N availability, and the relatively high apparent digestibility of the CP in the diet, we hypothesized that the alpacas in our study (high altitude) were very efficient at recycling urea, and thus it is likely that the synthesis of the microbial protein in C1 was optimized. Early studies have indicated that when alpacas are located at high altitudes, improved forage utilization occurs [6,36]. The nutrient requirements found in the present study for the adult male alpacas that were offered a high-quality diet should be revised to take into account the high productive balance of such animals in our experimental conditions (Peruvian Altiplano; altitude: 3704 m).

## 5. Conclusions

In conclusion, the total-tract apparent digestibility of the experimental diet, comprising oat hay and alfalfa pellets (in a ratio of 70:30 on a feed basis), remained relatively consistent among adult male alpacas, regardless of the offered level of intake. The apparent OMD *ad libitum* was 655 ± 51.1 g/kg DM, which is a value consistent with the reported range for alpacas on high-quality diets found in the literature. The slight elevation in DMI observed when the alpacas were fed at 50 g/kg BW^0.75^ was associated with a reduced consumption level of oat hay, indicating a decreased selectivity for this feed. In the present study, the ME intake at the maintenance level was estimated to be 347 ± 52.0 kJ/kg of BW^0.75^. The greater nutrient losses were attributed to the contents in the feces, irrespective of the level of intake. Although no differences were detected in the urinary excretion of N, the biological value of the dietary N (N retained/N absorbed) increased with the graded increase in the diet supply. A study aiming to investigate microbial N synthesis and saliva production might be helpful for a better understanding of the N recycling mechanisms displayed by alpacas when fed the same diet as the one offered in the present study. The assumption is made that the efficiency of dietary N usage was not influenced by the intake levels of these nutrients, as indicated by the C:N ratios in the feces.

## Figures and Tables

**Table 1 animals-13-03613-t001:** Chemical composition of the dietary ingredients (g/kg DM unless otherwise stated).

Nutrient	Oat Hay	Alfalfa Pellets	Diet ^1^
Dry matter	854	891	865
Ash	82.4	89.2	84.4
Crude protein	107	170	126
Neutral detergent fiber ^2^ (aNDF)	625	446	571
C	436	452	441
N	17.1	27.2	20.1
C-N ratio	25.5	16.6	21.9
Gross energy, MJ/kg DM	18.1	18.6	18.3
Est. Metabolizable Energy ^3^, MJ/kg DM	10.3	8.91	9.88

^1^ Diet is a 70:30 blend of oat hay and alfalfa pellets (% on a fed basis). Diet composition is calculated from dietary ingredients. ^2^ NDF assayed with a heat stable amylase and expressed inclusive of residual ash [25].^3^ Estimated for the total diet and calculated from tabulated values of individual feeds [26].

**Table 2 animals-13-03613-t002:** Feed intake, water consumption, and BW of adult male Huacaya alpacas (*n* = 5) when fed at two levels of intake of a diet containing oat hay and alfalfa pellets (70:30 ratio as a percentage on a fed basis).

Item	Maintenance				*Ad libitum*			
	Mean	s.d.	Minimum	Maximum	Mean	s.d.	Minimum	Maximum
Feed intake, g/d								
Oat hay	572	62.3	435	677	599	101	461	814
Alfalfa pellets	277	22.6	246	310	322	76.3	129	395
Total DMI	850	82.0	681	987	921	154	699	1210
DMI as a % of BW	1.32	0.068	1.21	1.44	1.41	0.166	1.10	1.60
DMI per-kg of BW^0.75^	37.4	1.83	34.2	39.8	40.0	4.89	31.1	47.2
OM	778	75.1	623	903	843	141	640	1107
CP	109	10.1	89	125	119	20.7	83	155
aNDF	481	47.6	382	561	518	85.4	396	685
GE, MJ/d	14.7	1.41	11.8	17.0	15.9	2.68	12.0	20.9
Selectivity ^1^, %								
Oat hay	91.2	6.10	79.3	99.3	75.5	8.95	58.5	92.1
Alfalfa pellets	98.9	2.52	90.2	100.0	90.7	19.6	36.9	100
Water consumption								
mL/d	2174	389	1210	3215	2746	683	1675	3920
as a % of BW	3.42	0.73	1.65	5.10	4.22	1.01	2.45	6.13
mL/kg of BW^0.75^	96.5	19.4	48.2	144	120	28.2	70.4	173
BW, kg								
Initial	64.4	6.50	54.0	73.5	65.3	6.85	55.0	75.5
Final	63.9	6.10	54.0	72.0	66.0	6.68	56.0	76.0
Average daily gain ^2^, g/d	−100	122	−300	0.0	140	54.8	100	200

^1^ Expressed as = (consumed feed/offered feed) × 100. ^2^ Calculated for each level of feeding as = (Final BW − Initial BW)/d.

**Table 3 animals-13-03613-t003:** Fecal output, urine production, and apparent total-tract digestibility of adult male Huacaya alpacas (*n* = 5) when fed at two levels of intake of a diet containing oat hay and alfalfa pellets (70:30 ratio as a percentage on a fed basis).

Item	Maintenance				*Ad libitum*			
	Mean	s.d.	Minimum	Maximum	Mean	s.d.	Minimum	Maximum
Fecal output, g/d								
Fresh basis	877	161	614	1200	1037	292	650	1922
DM	288	45.7	188	380	328	78.4	213	558
OM	256	43.8	163	331	293	76.7	184	522
CP	27.5	4.30	18.4	38.1	32.3	6.76	22.2	52.9
aNDF	210	35.5	140	263	241	62.9	150	427
Urine production								
Total, mL/d	429	119	244	604	445	174	160	700
As % of water cons.	20.8	7.33	10.3	37.7	17.7	8.72	4.82	35.2
Digestibility ^1^, g/kg								
DM	659	59.8	546	755	645	49.9	497	725
OM	669	56.6	576	772	655	51.1	486	725
CP	746	45.7	647	809	728	35.8	633	799
aNDF	561	80.6	420	705	538	70.8	314	635
GE	664	58.9	556	761	650	49.1	509	732

^1^ Apparent total-tract digestibility = (nutrient intake − fecal output)/(nutrient intake) × 1000.

**Table 4 animals-13-03613-t004:** Daily energy partitioning and balances of nutrients (carbon and nitrogen) of adult male Huacaya alpacas (*n* = 5) when fed at two levels of intake of a diet containing oat hay and alfalfa pellets (70:30 ratio as a percentage on a fed basis).

Item	Maintenance				*Ad libitum*			
	Mean	s.d.	Minimum	Maximum	Mean	s.d.	Minimum	Maximum
GE intake, kJ/kg of BW^0.75^	646	31.1	592	687	693	85.4	534	816
E Feces	217	37.7	149	273	243	49.2	166	369
E Urine	52.5	15.5	26.8	78.0	52.9	21.3	19.2	91.9
Est. CH_4_ energy ^1^	29.6	1.41	27.9	32.5	30.3	1.84	27.6	33.7
ME intake	347	52.0	238	419	366	70.4	245	505
Estimated CH_4_ emissions								
Total ^2^, g/d	17.0	0.67	15.6	18.1	17.6	1.27	15.7	19.9
*Y_m_* ^3^, % of GEI	4.59	0.270	4.19	5.22	4.42	0.424	3.77	5.16
CH_4_/DNDFI ^4^, g/kg	90.6	16.9	69.4	136	91.0	14.4	68.0	137
C balance ^5^, g/kg BW^0.75^								
C intake	16.5	0.80	15.1	17.5	17.7	2.17	13.7	20.8
C feces	5.55	0.97	3.75	6.85	6.23	1.30	4.20	9.55
C urine	0.327	0.097	0.174	0.486	0.344	0.137	0.125	0.566
N balance, g/kg BW^0.75^								
N intake	0.765	0.033	0.702	0.808	0.828	0.110	0.593	0.967
N feces	0.194	0.033	0.142	0.259	0.225	0.040	0.158	0.330
N absorbed	0.571	0.047	0.476	0.637	0.603	0.089	0.425	0.727
N urine	0.132	0.043	0.053	0.212	0.130	0.066	0.037	0.240
N retained	0.439	0.060	0.301	0.548	0.473	0.124	0.234	0.669
C:N ratios								
C:N intake	21.6	0.14	21.2	21.7	21.4	0.57	20.6	23.1
C:N feces	28.5	1.51	26.5	30.9	27.6	1.36	25.6	29.0
C:N urine	2.63	0.90	1.99	4.35	2.97	1.33	2.08	5.56

^1^ Estimated from prediction equation: Enteric CH_4_, g/d = 9.97 + 0.476 × GEI. Methane emissions were converted to energy unit equivalents using the constant 39.5 kJ/L CH_4_ [34]. ^2^ Total CH_4_ emissions are expressed in grams per day. The following conversion factor was used: 1 L = 0.716 g [35]. ^3^ Methane yield. ^4^ Rate of CH_4_ production per kg of digestible aNDF intake (DNDFI). The DNDFI is calculated as follows: aNDF intake × aNDF digestibility. ^5^ Carbon retention was not calculated as C gas losses from enteric fermentation (CH_4_ and CO_2_) were not measured in the present study.

## Data Availability

The data that support the findings of this study are available from the corresponding author, P.K.C.G., upon reasonable request.

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
