# Peer review of "Diet Digestibility and Partitioning of Nutrients in Adult Male Alpacas Fed a Blend of Oat Hay and Alfalfa Pellets at Two Levels of Intake"

_animals, 2023, doi:10.3390/ani13233613_

Round 1
Reviewer 1 Report
Comments and Suggestions for Authors
Dear Authors, as a result of my review, I have assessed the basic scientific content of your paper and suggest some language and wording corrections, along with a few suggestions. You can find my detailed comments on this issue in the pdf file. Below you will find a brief summary of these suggestions:
While the overall language quality of your paper is quite high, some sentences could use clearer and more precise wording. Definitions of terms could be made clearer for some readers.
Some of the sentences could be edited for more effective expression. When you examine the suggested changes, you will find that these improvements will improve fluency.
Consider adding more detail at some points to improve the strength of the scientific part.
I encourage you to take a look at the suggestions and make the necessary corrections to take your paper one step further and ensure the best quality. Please review your paper with these suggestions in mind. I believe that your article will become more effective and readable with your review and possible updates.

You can find detailed information on the English language in the pdf file.
Author Response
"Please see the attachment."

Reviewer 2 Report
Comments and Suggestions for Authors
This manuscript compares some indexes of production performance of alpaca feeding oat hay and alfalfa pellets, which has certain guiding value for production practice. However, the manuscript still needs to be improved. Specific suggestions are as follows:
1. The results of daily weight gain are not mentioned in the abstract.
2 In Table 1, the total energy value is not meaningful and is suggested to be expressed in terms of estimated metabolic energy.
3. The data in Table 2 and Table 3 are not fully displayed. Has a significant difference analysis been conducted?
4. Some indicators were measured in insufficient detail, such as water intake and urine collection.
5. How to calculate nitrogen absorption and retention in table 2? It is not mentioned in the text.
6. The feed intake of free feeding is higher than the maintenance level, and the digestibility of the maintenance level is higher than that of free feeding, why the daily gain of free feeding is so much higher than the maintenance level, it is recommended to discuss the reasons in detail.
Comments on the Quality of English LanguageThe logical relationship described in the manuscript needs to be perfected.
Author Response
"Please see the attachment."

Reviewer 3 Report
Comments and Suggestions for Authors
Guillen et al. evaluated the effects of two levels of dry matter intake on a metabolic body weight basis level of intake (40 and 50 g of DM per kg of BW0.75, respectively) on apparent diet digestibility and partitioning of nutrients of alpacas fed a blend of oats hay and alfalfa pellets. This study helps to understand the relationship between diet digestibility and intake in alpacas. The experiment is well designed, but the manuscript requires some corrections.
1. in “Abstract”, the abbreviations must be defined at their first mention.
2.in “Materials and Methods”, why was the ratio of oat hay to alfalfa pellets determined to be 70:30?
3. line 138, “at 0600 h” changed as “at 06:00 h”.
4. line 143, “at 0630 h” changed as “at 06:30 h”.
5. in “Results”, table 3 is not described.
6. Table 2-Table 4 are not complete, please replace.
Author Response
"Please see the attachment."

Round 2
Reviewer 1 Report
Comments and Suggestions for Authors
Dear authors, thank you for resubmitting your revised manuscript. I am happy to note that authors have worked really hard to improve the manuscript and all the comments raised by me were not only answered but also implemented to improve the manuscript. Hence, I recommend the manuscript to be published in this journal. Thanks for hard working.
Reviewer 2 Report
Comments and Suggestions for Authors
The author has finished revising the manuscript according to the revised opinions and agrees to publish it on animals.
Reviewer 3 Report
Comments and Suggestions for Authors
Accept in present form.